# Impact of 'HIV-related stigma-reduction workshops' on knowledge and attitude of healthcare providers and students in Central India: a pre-test and post-test intervention study

Anna Machowska,[1] Babu Lal Bamboria,[2] Courtney Bercan,[1] Megha Sharma [iD] [1,3]

AM and BLB contributed equally.

AM and BLB are joint first authors.

¹Department of Global Public Health, Karolinska Institutet, Stockholm, Stockholm County, Sweden
²Department of Medicine, R. D. Gardi Medical College, Ujjain, Madhya Pradesh, India
³Department of Pharmacology, R. D. Gardi Medical College, Ujjain, Madhya Pradesh, India

**Correspondence to**
Dr Megha Sharma;
meghasharma27@rediffmail.com

## ABSTRACT

**Introduction** People living with HIV and AIDS (PLWHA) experience stigma and discrimination throughout their life. The consequences of stigma and discrimination are severe when enacted by healthcare providers (HCPs), and result in a delay in or poor adherence to treatment. Studies have demonstrated the presence of stigmatising behaviours among HCPs, yet only a few have presented the impact of interventions on the knowledge and attitude of practising (clinicians and nurses) and future (students) HCPs.

**Objectives** To evaluate knowledge, attitudes and infection risk perceptions related to HIV/AIDS among practising and future HCPs in central India. In addition, the impact of the 'HIV-related stigma-reduction workshop' using a pre-intervention and post-intervention study design was evaluated.

**Settings and participants** The study was conducted in 2016 at two tertiary-care hospitals and three healthcare colleges in central India. Overall, 650 HCPs (75 clinicians and 211 nurses) and students (205 medical, 123 nursing and 36 Diploma in Medical and Laboratory Technology (DMLT)) voluntarily participated in the study.

**Intervention** The educational intervention comprised of training in epidemiology, the transmission and treatment of the HIV infection, the rights of PLWHA, the duties of HCPs and the use of standard precautions.

**Results** At pre-intervention, DMLT students had the lowest and clinicians the highest knowledge scores (24% and 45%, respectively). The stigmatised attitude was reflected in all groups, the lowest among clinicians (21%) and the highest among DMLT students (34%). Improvement in the post-intervention knowledge scores was the highest in medical students (36%) and the lowest among clinicians (16%). The participants' attitudes improved between 3% and 17% across all groups.

**Conclusions** Significant post-intervention improvements were seen in both knowledge and attitudes in all groups. Students had a higher tendency to improve than HCPs. Further long-term studies are needed to evaluate the sustainability of the improvements in knowledge and attitudes of the participants.

## Strengths and limitations of this study

► A context-specific educational intervention was implemented to improve knowledge and attitudes of practicing and future HCPs towards people living with HIV and AIDS.

► The study was conducted at two hospitals and three educational institutes where the participants work and/or study. Thus, we may observe a social desirability bias, resulting in the underestimation of stigma and discrimination among the participants.

► The end goal of any educational intervention is to reduce stigmatising behaviours, but behaviour change is a long-term process that is complex to capture and quantify; thus, attitudes were captured as a proxy.

► The post-intervention questionnaire was completed immediately after the intervention due to feasibility reasons. Thus, we cannot claim the sustainability of the positive effect of the intervention.

► Additional follow-ups would provide stronger evidence of the long-term positive impact of the study on attitudes and knowledge.

## INTRODUCTION

The HIV and AIDS epidemic is a global challenge. At the end of 2017, it was estimated that 36.9 million people were either infected by HIV or living with AIDS (people living with HIV and AIDS; PLWHA) globally. Of these, about 22 million people could access antiretroviral therapy (ART) and 940 000 people had died from HIV-related illnesses.[1 2] The United Nations' 90-90-90 targets launched in 2014 have become a central pillar of global efforts to end the HIV/AIDS epidemic. The aim is to diagnose 90% of all people who are HIV positive, provide ART for 90% of those diagnosed and achieve viral suppression for 90% of those treated by 2020. The programme aims to 73% of people with HIV achieving viral suppression, a crucial step in

ending the AIDS epidemic, by 2030.[3] Stigma, specifically HIV/AIDS-related stigma, is one of the key aspects of the HIV/AIDS epidemic. This stigma can be described as a socially shared perception of the devalued status of PLWHA.[4] The presence of stigma and discrimination against PLWHA at various societal levels fuels the HIV epidemic, and the consequences are more severe if the stigma is present among healthcare providers (HCPs).

HCPs are the core and integral part of the healthcare delivery system, and play a pivotal role in creating a positive environment for better health outcomes for PLWHA and preventing viral transmission.[5] However, inadequate knowledge about the HIV infection epidemiology among HCPs gives rise to fear and misconceptions about transmission of the infection.[6–8] Fear and misconceptions alter attitudes and enhance the development of stigma and discrimination towards PLWHA. Some of the major consequences of this are delays in seeking healthcare, less timely diagnosis, an unwillingness to disclose the HIV status in the community and while seeking healthcare and a delay in or non-compliance to the ART. Not disclosing an HIV status puts society at higher risk of infection transmission, contributes to increased morbidity and mortality of PLWHA, increases healthcare costs and places a burden on the healthcare sector overall.[9–11] Therefore, in addition to the diagnosis and treatment, provision of discrimination-free and stigma-free healthcare is a prerequisite for achieving the sustainable development target designed to end the epidemic.[12]

In 2016, with a prevalence of 2.1 million PLWHA, India was ranked third in the world after South Africa and Nigeria.[13 14] Although the prevalence of PLWHA varies between states in the country, the threat of the infection spreading is high throughout the country due to high levels of interstate and interdistrict migration.[15] Global organisations and governments in countries with a high infection risk, including India, have prioritised stigma-reduction through various intervention programme.[16] In India, the National AIDS Control Organisation advocates the prohibition of social discrimination against PLWHA.[14 16] The organisation has identified seven districts with high HIV prevalence; Ujjain is an administrative unit that includes four of these high-risk districts (Neemuech, Mandsour, Ujjain and Ratlam).[17 18] However, the data available in various parts of the country show that PLWHA have experienced stigma and discrimination from the community and HCPs.[19 20]

Educational interventions intended to improve knowledge of HIV/AIDS are effective in reducing stigma among HCPs and improving healthcare delivery.[12 21–25] Stigma-reduction intervention studies are, in general, time-consuming and resource-intensive, and most of them have mainly focused on nurses and nursing students, and not on clinicians and healthcare students.[10 22–24] However, medical, laboratory technician and nursing students are future HCPs which means it is important to develop appropriate interventions to improve their knowledge and reduce their HIV-related stigma while targeting

both practising (clinicians and nurses) and future (the students) HCPs.[26] However, we know little about HIV/AIDS-related knowledge, attitudes and infection risk perceptions of practising and future HCPs in India.

### Objectives
To evaluate knowledge, attitudes and infection risk perceptions related to HIV/AIDS among HCPs and healthcare students in the Ujjain district of central India. Furthermore, to evaluate and compare the impact of the 'HIV-related stigma-reduction workshop' intervention on the participant groups using a pre-intervention and post-intervention study design.

### Methods
This intervention study was conducted between January 2016 and August 2016 at three colleges of healthcare courses and two tertiary-care hospitals in Ujjain district in Madhya Pradesh state in India. One of the hospitals is associated with a medical college, a nursing college and a Diploma in Medical and Laboratory Technology (DMLT) college and is referred to as the teaching hospital (TH). An ART centre with specialised healthcare facilities for the PLWHA is located at the TH. Another participating hospital is referred to as the non-teaching hospital (NTH). In this study, the students of Bachelor of Medicine, Bachelor of Surgery, are referred to as medical students.[27]

### Patient and public involvement
There was no direct involvement of patients or the public in the present study.

### Participants
The participants were practising HCPs, that is, clinicians and nurses, and future HCPs, that is, students of medical, nursing and DMLT courses enrolled in respective institutes in Ujjain district of central India.

### Sampling strategy
All groups of the HCPs and students were invited to participate in the workshop through various means such as leaflets posted at the notice boards, announcements during lectures or meetings. The leaflets were written in Hindi and English to offer equal chances for all potential participants to learn about and attend the workshop to enhance their HIV/AIDS-related knowledge and ways to stay protected at the workplace. The workshops were conducted at respective colleges and hospitals to provide an equal opportunity to participate.

### Inclusion and exclusion criteria
Participation was voluntary, and the participant could leave the session at any time without experiencing any repercussions. The participants who could not attend or complete both the pre-workshop and post-workshop sessions were excluded from the analysis (figure 1).

### Structure of the workshops
The intervention was designed as an educational workshop that suited best the local context. All workshops

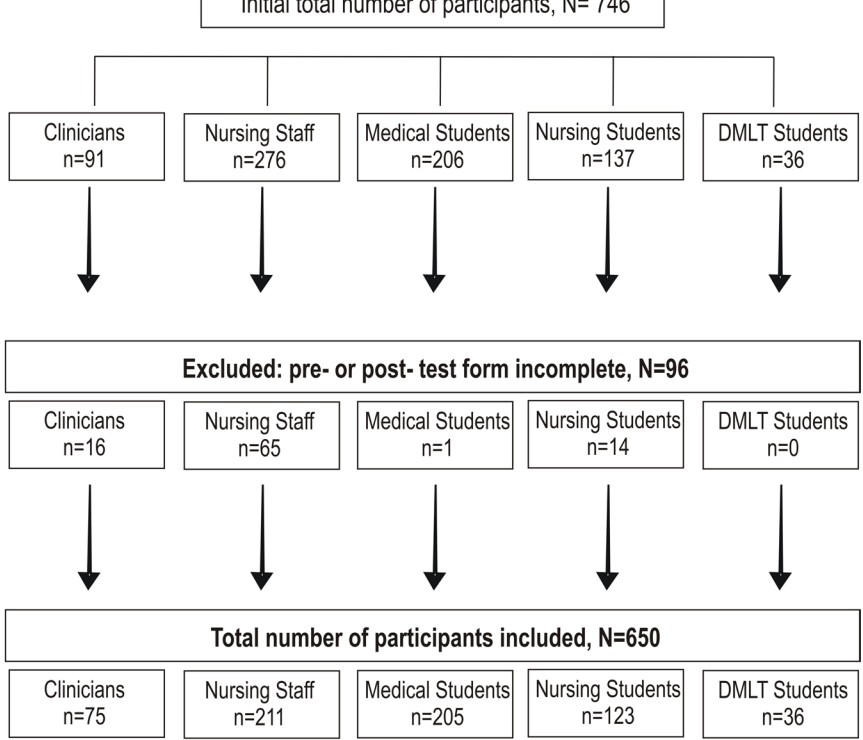

**Figure 1** Inclusion and exclusion criteria of the study participants. DMLT, Diploma in Medical and Laboratory Technology.

were conducted by the same facilitators: a clinician and an officer-in-charge of the ART centre at the TH (BLB), and the team leader of hospital infection prevention and control team at the TH and the NTH (MS). In total, 13 workshops were conducted. Participants were divided into smaller groups as follows: clinicians (two groups), nurses (five groups), medical students (three groups), nursing students (two groups) and DMLT students (one group) to ensure that they feel comfortable in the learning environment.

### Content of the workshops

The workshop was 2 hours long and divided into three sessions: the first session consisted of an animated video demonstrating the HIV infection epidemiology, the mechanism of viral transmission, diagnostic testing, treatment options, how the ART medicines work and ethical issues related to PLWHA and ART. This was followed by a second session that involved a PowerPoint presentation focusing on topics such as the rights of PLWHA, the duties of HCPs and predictable consequences due to presence of stigma and discrimination against PLWHA. The third session was more interactive and dynamic. Here, information about universal precautions and infection prevention and control measures was provided, describing the importance of the use of standard precautions, including hand hygiene in the daily work-routine. It was explained how and when to use gowns, gloves, face shields, masks and goggles, and the availability of postexposure prophylaxis in the study settings and the use of an in-house prepared alcohol-based hand rub (ie, 'Micro *Kavach*') was discussed.[28] This was followed by the demonstration

and practice of the WHO-recommended hand hygiene techniques using 'Micro *Kavach*'.[28] A period of 15 to 20 minutes was assigned for an open discussion. The participants were asked to complete the questionnaire twice: once before the workshop (pre-intervention) and then after the break (post-intervention).

### Data collection tool and analysis

The context-specific questionnaires were developed locally and were used as pre-intervention and post-intervention data collection tools. The questionnaires were pretested in each potential participant groups (not included in this study). After receiving feedback, the questionnaires were modified and finalised, and brief educational intervention was designed considering the time constraints among the participants.[29] The questionnaires were formatted in both English and Hindi to facilitate the understanding of the questions by all participant groups. The questionnaires were divided into the five following sections[1]: participants' demographic information,[2] work experiences,[3] HIV/AIDS-related knowledge,[4] attitudes towards PLWHA,[5] risk perception and willingness to provide care to the PLWHA and patients with hepatitis B.

The responses related to the knowledge and attitude questions were scored in the following manner: each correct response in the knowledge section received a score of one, and each incorrect response received a score of zero. Scores of each individual were converted to the percentage of correct responses. Like in previous publications in this field; the levels of knowledge were categorised into 'low' for the participants who scored ≤50%,

'moderate' for those who scored between 51% and 74%, and 'high' for those who scored ≥75%.[30]

Responses to the questions related to participants' attitude towards PLWHA were classified into three categories: positive (non-stigmatising), unsure and negative (stigmatising).[31] A positive response was given a score of one and the responses where the participants were unsure or, where the responses indicated a negative attitude including fear, stigma or intent to discriminate the PLWHA, were given a score of zero.[32]

Scores of each individual were added and converted to the percentage of correct responses. Responses, where the missing data was higher than 50%, were excluded from the analysis. Risk perception questions were categorical and dichotomised according to the outcome of interest, for example, (a) I am willing or not willing to care for a patient or (b) I am at high risk or not at high risk of HIV exposure at the workplace.

### Statistical analysis

The data on demographic and basic HIV-related information was presented as mean and standard deviation (SD) for continuous variables and as percentages and frequencies for categorical variables. The knowledge, attitudes and work environment variables were presented as median with interquartile range (IQR). The difference between the pre-intervention and post-intervention scores were analysed using a Wilcoxon signed-rank test for continuous variables and McNemar's test for categorical variables. The score difference was calculated by subtracting the pre-intervention (baseline) score from the post-intervention score and represents the percentage unit increase or decrease concerning the baseline score. As these scores were not normally distributed, a Kruskal-Wallis test was conducted to determine a statistically significant difference among the participant groups in pre-intervention and post-intervention scores as well as the difference of scores. Post-hoc Dunn's tests were conducted to evaluate the relationship between the groups. The statistical significance was set at the level of p<0.05. The data were entered in Epi info software (V.3.1) and Microsoft Excel and statistical analysis was performed using the Statistical Package for the Social Sciences (SPSS V.22, Chicago, Illinois, USA) and STATA V.15.0 (Stata, College Station, Texas, USA) software.

### RESULTS
#### Baseline characteristics

In total, 920 people were invited to take part, and 746 (81%) voluntarily participated in the study. Of these, 650 (87%) attended all three sessions of the workshop and completed the questionnaire twice; 56% were students and 44% were practising HCPs. Thus, data from these 650 participants was included in the analysis (figure 1).

The mean age of the participants was 23 years, the majority were of the Hindu faith and 56% were males table 1. The proportion of participants who had received HIV/AIDS-related training previously varied from 4%

in the clinicians' group to 78% in the DMLT students' group. Only 2% of medical students stated that they had worked with PLWHA before. A high proportion of participants in each group reported experiencing accidental exposure to HIV or PLWHA. A range of 9% to 20% of the participants admitted discriminating against PLWHA in workplaces. Six (3%) medical students and 11% of DMLT students admitted that they had refused to provide care for PLWHA during their scheduled clinical visits, while between 17% and 28% of participants thought that their colleagues discriminated against PLWHA (table 1).

### Impact of the intervention on knowledge

The pre-intervention knowledge score, presented as the percentage of correct responses, demonstrated the low overall level of knowledge, with the poorest in DMLT students (22%) and the highest among the clinicians (45%, table 2). The post-intervention knowledge scores ranged from 48% in the nurses' group to 64% in the medical students' group. The post-intervention knowledge scores were significantly higher than the pre-intervention knowledge scores (p<0.001, table 2), and almost all groups reached a moderate level of knowledge. The difference between the knowledge scores of all study groups was statistically significant (p<0.001). The knowledge scores of both medical and nursing students improved more than their respective practising HCPs. A pairwise comparison showed that at the pre-intervention stage, the clinicians had the highest knowledge level of all the groups (p<0.001, table 3). However, at the post-intervention stage, the medical students presented higher knowledge levels than the clinicians, while the knowledge of the nurses was the lowest.

### Impact of the intervention on attitude

Evidences of a stigmatising attitudes were observed among all respondent groups. The lowest level of stigmatising attitudes at pre-intervention was found in the clinicians' group, where 79% responded to the questions without stigma. On the other hand, in the DMLT students' group, only 66% of responses presented a non-stigmatising attitude (table 2). At post-intervention, the percentages of non-stigmatising responses were higher, ranging from 79% (lowest) among the nurses to 84% in the medical students' group. The attitude improved significantly across all groups, with a three percent increase in the clinician group and 17% in the DMLT students' group (p<0.001). Overall, the intervention resulted in an improvement in stigma levels of up to 21%, as shown in the nurses' group. A pairwise comparison showed that medical students had the greatest improvement in their attitude (table 4).

### Impact of the intervention on infection risk perception

Thirty per cent of the medical students and 67% of the clinicians perceived themselves to be at high risk of HIV infection in their workplace at the pre-intervention stage. The willingness to care for PLWHA was relatively high at the pre-intervention stage, ranging from 72% among

**Table 1** Baseline demographic information (part A) and experiences (part B) of the participants

| | Clinicians (n=75) | Nurses (n=211) | Medical students (n=205) | Nursing students (n=123) | DMLT students (n=36) |
|---|---|---|---|---|---|
| Part A: Baseline demographic information | | | | | |
| Age, mean (SD) | 28 (4) | 25 (6) | 20 (1) | 21 (3) | 20 (2) |
| Sex, n (%) | | | | | |
| Male | 30 (40) | 148 (70) | 109 (53) | 51 (41) | 28 (78) |
| Female | 45 (60) | 63 (30) | 96 (47) | 72 (59) | 8 (22) |
| Married, n (%) | 21 (28) | 70 (33) | – | 11 (9) | 35 (97) |
| Religion, n (%) | | | | | |
| Hindu | 62 (83) | 192 (91) | 183 (89) | 120 (98) | 31 |
| Muslim | 2 (3) | 5 (2) | 6 (3) | 3 (2.4) | 4 |
| Christian | 2 (3) | 10 (5) | 1 (0.5) | – | – |
| Other | 9 (12) | 4 (2) | 15 (7) | – | 2 |
| Have you ever attended any HIV/AIDS-related training? Yes, n (%) | 3 (4) | 39 (18) | 63 (31) | 28 (23) | 28 (78) |
| Part B: Experiences of participants | | | | | |
| Have you ever cared for PLWHA before? Yes, n (%) | 37 (49) | 121 (57) | 4 (2) | 88 (72) | 18 (50) |
| If yes, how many times have you cared for PLWHA? | | | | | |
| Less than 10 times | 29 (78) | 59 (49) | 2 (50) | 61 (69) | 5 (28) |
| Less than 50 times | 2 (5) | 27 (22) | – | 14 (20) | 8 (44) |
| More than 100 times | 1 (3) | 19 (16) | – | – | 5 (28) |
| Cannot recall the numbers/too many | – | 16 (13) | 2 (50) | 13 (15) | – |
| §Have you ever had an accidental exposure with body fluid of PLWHA, at work? Yes, n (%)* | 4 (11) | 24 (20) | 3 (75) | 14 (16) | 1 (6) |
| Have you ever refused to care for PLWHA? Yes, n (%) | 6 (8) | 18 (9) | 6 (3) | 6 (5) | 4 (11) |
| Have you discriminated against PLWHA at the workplace? Yes, n (%) | 7 (9) | 42 (20) | 22 (11) | 13 (11) | 5 (19) |
| Do you think your colleagues discriminate against PLWHA? Yes, n (%) | 21 (28) | 45 (21) | 51 (25) | 28 (23) | 6 (17) |
| Is your family aware that you ought to care for PLWHA patients as well? Yes, n (%) | 49 (65) | 134 (64) | 127 (62) | 100 (81) | 16 (72) |
| Is there any family pressure regarding caring for PLWHA? Yes, n (%) | 15 (20) | 46 (22) | 16 (8) | 15 (12) | 8 (22) |
| Do you have any friend or family who is identified as PLWHA? Yes, n (%) | 3 (4) | 16 (8) | 3 (2) | 5 (4) | 2 (6) |
| There is enough equipment to maintain universal precautions at workplace. Yes, n (%) | 20 (27) | 126 (60) | 79 (39) | 84 (68) | 19 (53) |
| Have you ever had any accidental exposure to HIV through PLWHA at work? Yes, n (%) | 4 (5) | 24 (11) | 3 (1) | 14 (11) | 1 (3) |

Yes, n (%) is the proportion of the participants who responded 'Yes' to the questions asked.
The variables where totals do not add up to 100%, the remaining percentage of the participants did not respond to that question.
*Proportion calculated from patients who confirmed that they have ever cared for PLWHA.
DMLT, students of Diploma in Medical and Laboratory Technology; PLWHA, people living with HIV/AIDS; Medical students/ MBBS students, students of Bachelor of Medicine, Bachelor of Surgery.

the nurses to 94% among the DMLT student group. The proportion of the participants who were willing to care for people living with hepatitis B was similar to those willing to care for PLWHA.

After the workshop, the medical and nursing students reported a significant increase in their awareness of the risk of HIV infection compared to their statements at pre-intervention. There was a significant increase in the post-intervention willingness to care for PLWHA in the nursing group, and a non-significant decrease in the DMLT and nursing students' group (table 5).

## DISCUSSION

The present study evaluated the effect of 'HIV-related stigma-reduction workshops' and showed that there was a

**Table 2** Pre-intervention and post-intervention scores of knowledge and attitude among the participant groups

**Knowledge**

| | Clinicians (n=75) % (IQR) | Nurses (n=211) % (IQR) | Medical students (n=205) % (IQR) | Nursing students (n=123) % (IQR) | DMLT students (n=36) % (IQR) |
|---|---|---|---|---|---|
| Percentage of correct responses | | | | | |
| Pre-intervention, median (IQR) | 45 (15) | 30 (19) | 29 (14) | 30 (19) | 22 (14) |
| Post-intervention, median (IQR) | 59 (14) | 48 (26) | 64 (21) | 59 (26) | 52 (17) |
| Difference | 14* | 18* | 35* | 29* | 30* |
| **Attitude** | | | | | |
| Percentage of responses without stigma | | | | | |
| Pre-intervention, median (IQR) | 79 (18) | 74 (15) | 74 (18) | 76 (12) | 66 (35) |
| Post-intervention, median (IQR) | 82 (11) | 79 (12) | 84 (16) | 82 (9) | 83 (14) |
| Difference | 3* | 5* | 10* | 5* | 17* |

*Statistically significant differences with significance level p<0.05.
DMLT, students of Diploma in Medical and Laboratory Technology; IQR, Interquartile range; Medical students/ MBBS students, students of Bachelor of Medicine, Bachelor of Surgery.

significant improvement in HIV/AIDS-related knowledge in all groups who participated in the intervention, irrespective of their inadequate baseline knowledge. Our study highlighted the fact that stigmatising attitudes were widespread among practising HCPs and students of various healthcare courses. However, stigmatising attitudes and discrimination that led to an unwillingness to provide medical care to PLWHA were significantly reduced among the participants after the intervention. The student groups showed more receptiveness to improve their knowledge and attitudes towards PLWHA than the practising HCPs.

### Knowledge

Low knowledge levels can perpetuate myths and misunderstandings that extend to baseline beliefs and stigmatising attitudes. The present study showed that while the baseline

HIV knowledge of both the practising HCPs and the healthcare students was inadequate, there was a significant knowledge improvement in all participant groups after the intervention. These findings are comparable with results from a cross-sectional survey conducted among practising HCPs and healthcare students in Uttarakhand, India.[32] That study also found low levels of HIV-related knowledge among HCPs and highlighted the need for improved HIV-specific education. Our study echoed their finding that clinicians had more knowledge than the other groups. This is understandable, as physicians are likely to receive more HIV-specific training than other types of HCPs.[33]

In our study, the overall post-intervention knowledge scores significantly increased in all groups, and particularly so in the students' groups. This was similar to the

**Table 3** Pre-intervention and post-intervention pairwise comparison of knowledge between the participant groups

| Groups compared | Pre-intervention knowledge-adjusted significance | Post-intervention knowledge-adjusted significance | Pre-intervention and post-intervention knowledge-adjusted significance |
|---|---|---|---|
| DMLT students–nursing students | <0.05* | 0.068 | 1.000 |
| DMLT students–medical students | <0.01* | <0.001* | 0.629 |
| DMLT students–clinicians | <0.001* | 0.116 | <0.001* |
| DMLT students–nurses | 0.059 | 1.000 | 0.234 |
| Medical students–clinicians | <0.001* | 0.051 | <0.001* |
| Nursing students–clinicians | <0.001* | 1.000 | <0.001* |
| Nursing students–medical students | 1.000 | <0.05* | 0.934 |
| Nurses–nursing students | 1.000 | <0.001* | <0.001* |
| Nurses–clinicians | <0.001* | <0.001* | <0.05* |
| Nurses–medical students | 1.000 | <0.001* | <0.001* |

Significance values have been adjusted by the Bonferroni correction for multiple tests.
*Statistically significant differences with significance level p<0.05.
DMLT, students of Diploma in Medical and Laboratory Technology; Medical students/ MBBS students, students of Bachelor of Medicine, Bachelor of Surgery.

 Machowska A, *et al. BMJ Open* 2020;**10**:e033612. doi:10.1136/bmjopen-2019-033612

**Table 4** Pre-intervention and post-intervention pairwise comparison of attitudes between the participant groups

| Groups compared | Pre-intervention attitude-adjusted significance | Post-intervention attitude-adjusted significance | Pre-intervention and post-intervention attitude-adjusted significance |
|---|---|---|---|
| DMLT students–nursing students | 0.184 | 1.000 | 0.147 |
| DMLT students–medical students | 0.479 | 0.614 | 1.000 |
| DMLT students–clinicians | <0.01* | 1.000 | 0.134 |
| DMLT students–nurses | 0.627 | 1.000 | 0.439 |
| Medical students–clinicians | <0.05* | 1.000 | <0.01* |
| Nursing students–clinicians | 0.252 | 1.000 | 1.000 |
| Nursing students–medical students | 1.000 | 0.173 | <0.01* |
| Nurses–nursing students | 1.000 | 0.614 | 1.000 |
| Nurses–clinicians | <0.05* | 0.092 | 1.000 |
| Nurses–medical students | 1.000 | <0.001* | <0.05* |

Significance values have been adjusted by the Bonferroni correction for multiple tests.
*Statistically significant differences with significance level p<0.05.
DMLT, students of Diploma in Medical and Laboratory Technology; Medical students/ MBBS students, students of Bachelor of Medicine, Bachelor of Surgery.

effect of educational interventions on HIV knowledge in two single-centre studies involving nursing students in Bengaluru and nurses in Delhi, India.[22 24] In Bengaluru, the intervention was divided into two 1-hour sessions, while in Delhi it involved a 2-day training programme.

We hypothesised that there would be a greater post-intervention increase in knowledge among the clinicians' group than the other groups. However, the students' groups showed significant knowledge gains and the clinicians showed the lowest increase. Theories suggest that knowledge is associated with baseline beliefs and attitudes, and depends on the readiness of the learners and their willingness to integrate the change into their behaviour.[34 35] As per the educational theory, the recognition of gap in knowledge and skills, or the need for behaviour change, is essential for making the desired change.[34 35] The student groups, being in the learning phase of life, may have identified their gaps in knowledge, thus making them more receptive to the training than the other groups,[34 35] while the clinicians had adequate knowledge at pre-intervention and thus had less scope to improve their knowledge at the post-intervention stage.

## Attitude

Stigmatising attitudes towards PLWHA can feed into a lack of desire or willingness to learn more about the disease. The present study confirmed our hypothesis that stigmatising attitudes are widespread among practising HCPs and healthcare students in the region. Our pre-intervention stigma levels were higher in healthcare students and lower in practising HCPs, which is consistent with a cross-sectional study of HCPs in Uttarakhand, India.[33] Professional experience with PLWHA is a predictor of more positive attitudes towards them.[14 27]

**Table 5** Risk perception reported by the participants in the pre-intervention and post-intervention questionnaires

| Questions | Time points | Nurses (n=211) n (%) | Clinicians (n=75) n (%) | Nursing students (n=123) n (%) | Medical students (n=205) n (%) | DMLT students (n=36) n (%) |
|---|---|---|---|---|---|---|
| Do you think you are at high risk for HIV/AIDS at work? Yes, n (%) | Pre-intervention | 71 (34) | 50 (67) | 39 (32) | 61 (30) | 22 (61) |
| | Post-intervention | 81 (38) | 51 (68) | 86 (70)* | 82 (40)* | 17(47) |
| Are you willing to care for: | | | | | | |
| (a) Patients with hepatitis B (yes), n (%) | Pre-intervention | 174 (83) | 64 (85) | 112 (91) | 155 (76) | 28 (78) |
| | Post-intervention | 192 (91)* | 65(87) | 118 (96) | 196 (96)* | 32 (89) |
| (b) PLWHA (yes), n (%)† | Pre-intervention | 151 (72) | 65 (87) | 114 (93) | 179 (87) | 34 (94) |
| | Post-intervention | 175 (83)* | 66 (88) | 110 (89) | 183 (89) | 30 (83) |

Yes, n (%) is the proportion of the patients who responded 'Yes' to the questions asked.
*Statistically significant differences with significance level p<0.05.
†The question was also classified as an attitude question.
.DMLT, students of Diploma in Medical and Laboratory Technology; Medical students/ MBBS students, students of Bachelor of Medicine, Bachelor of Surgery; PLWHA, people living with HIV/AIDS.

The higher levels of stigmatising attitudes found among participants (students) in our study could be due to incomplete or inadequate knowledge, and relatively less experience of working with PLWHA.

Studies show that working on empathy-building techniques and including PLWHA testimonies in interventions can improve outcomes for HCPs and contribute to decreasing the stigma and discrimination towards PLWHA.[36 37] Recent studies demonstrate that small efforts such as support in the workplace, HCP-friendly policies and displaying stigma-reduction messages can change HCPs' attitudes towards PLWHA.

Similar to the knowledge scores, the student groups showed larger improvements in their stigmatising attitudes than the working professionals. We must acknowledge, however, that although improvements in knowledge were seen in all groups, increased knowledge is not typically enough to affect changes in a person's attitudes and the baseline beliefs that contribute to stigma and discrimination. These changes are the result of a complex interplay between knowledge, baseline beliefs and attitudes that affects the readiness of the learners and their willingness to integrate the change into their behaviour.[10 34 35]

### Risk perception

Another highlight of our study was the willingness to care for PLWHA shown by the majority of HCPs and students at the pre-intervention stage. Our findings are in line with a study conducted in India by Kermode *et al* that showed a relatively high willingness of HCPs to care for PLWHA despite the presence of stigmatising attitudes.[7] However, there are few studies that assess risk perception prior to and after a stigma-reduction intervention.

Medical and nursing students significantly improved their perception of HIV/AIDS risk in the post-intervention test. The same receptiveness to learning that was shown by the student groups in our study is likely to be the reason their knowledge in this area also improved. It must be said, however, that educational interventions may have a limited effect on risk perception if appropriate personal protective materials are not available in the workplace. This points to the necessity to create structural and policy changes that contribute to a safer work environment in conjunction with educational interventions in order to both reduce perceived and actual workplace risk.[6 7] Further studies are needed to determine the factors affecting HCPs' willingness to care for PLWHA.

Our study is based on the call for new research that promotes the development of knowledge-building and stigma-reduction interventions to target a broad range of HCPs that PLWHA will encounter in their healthcare journeys.[38 39] The scaling-up of HIV stigma-reduction efforts lacks focus throughout the world, and especially in countries with a high HIV prevalence like India. The major challenge is implementation feasibility of the interventions mainly due to limitations in time commitment, cost and available resources. However, the intervention adapted in present study is low resource, low cost and less time consuming with increased implementation feasibility.

The number and duration of educational sessions does not necessarily correspond to greater increases in knowledge, and shorter interventions may suffice to bring participants to an adequate level of knowledge.[38 39] Researchers have used various time lengths to assess their intervention designs, and thus it is not appropriate to conclude there is a relationship between the length and number of workshops and the level of change.[22 24 36 40] However, it seems that even a short workshop can produce positive results, as seen by the progress of the students' groups in our study.

To the best of our knowledge, this is the first study that presents and compares pre- and post-intervention knowledge and stigma in terms of attitudes towards PLWHA, among various groups of practising and future HCPs. The context-specific low-cost educational intervention was feasible and appreciated by most of the HCPs during both the pilot testing and the real study. Our intervention is brief and requires relatively few resources and little time, which is important in settings where HCPs work. These features increase the generalisability and feasibility of adopting the intervention design and applying it in other similar settings where HCPs are burdened by a heavy workload.

This study shows that an educational and skills-based intervention, even if only for 2 hours, can be effective in increasing knowledge and reducing stigmatised attitudes. Therefore, recurrent training and educational workshops might have a sustainable effect of improved knowledge and reduced stigmatised attitudes and behaviour towards PLWHA. Finally, the present study fills a knowledge gap by providing a better understanding of knowledge, attitudes and risk perceptions of various HCPs and healthcare students in India and globally.

Like any study, this methodology must be seen in the context of its limitations. The major consideration of our study is that the post-intervention questionnaire was completed after a short break following the intervention. Thus, we could not validate whether the reduction in stigmatising beliefs was sustained over time. The end goal of the intervention was to reduce stigmatised behaviour. However, behaviour change is a long-term process and is complex to capture and quantify; thus, we were restricted to capturing attitudes as a proxy. Although the present one-time post-intervention data collection design presents a strong association between the intervention and the results, additional follow-ups, for example, at 4 weeks, 6 months and 1 year, would provide stronger evidence for the positive impact of the intervention on attitudes and knowledge than the current design.

Given that the study was conducted in workplaces, the presence of social desirability bias in participants' stigma-related responses cannot be ruled out. Therefore, it is possible that the rate of stigma presented in the study is underestimated. The voluntary participation, with an emphasis on increasing knowledge, and the anonymous

questionnaire completion, might have motivated HCPs to participate in the study.

## CONCLUSIONS

Combating stigma is a key priority in achieving success in epidemic control and providing quality care to PLWHA. The findings from this study show that the HIV-related stigma-reduction workshops that are brief, feasible and relatively low-resource interventions, can successfully increase knowledge and reduce stigma in both HCPs and students. Considering the generalisability and feasibility of the intervention design, repetitive contextualised workshops with extended follow-ups are recommended to evaluate their impact and assure sustainable effect. We propose incorporating of more practical and integrated educational training in the teaching curricula of all healthcare courses. This will benefit the HCPs, PLWHA and the community. Moreover, in future studies, we recommend the inclusion of a qualitative component and behavioural assessment to understand participants' attitudes and willingness to change stigmatising behaviour towards PLWHA.

**Acknowledgements** We thank all study participants, the management and staff of the hospitals and colleges for facilitating and supporting the study. We are thankful to Mr Amit Pawar for assisting in data collection and data management.

**Contributors** MS and BLB perceived the study. MS and BLB designed the questionnaire and the content of the interventions. MS and BLB conducted the training sessions and collected the pre-intervention and post-intervention data. AM, CB and MS performed the analysis. AM, MS and CB drafted the article. AM, MS, BLB and CB gave critical comments to modify the manuscript. All authors gave final approval to the present version for publication.

**Funding** This work was supported by Swedish Research Council grant number 2017-01327.

**Competing interests** None declared.

**Patient consent for publication** Not required.

**Ethics approval** Ethical approval was granted by the ethics committee of R. D. Gardi Medical College, Surasa, Ujjain, India with letter numbers 311/17-07-2013 and 249/23-01-2013. Verbal and written consents were obtained from each participant before the study. Participants were assured by the researchers for full confidentiality. Participants were encouraged to respond to the questionnaires fearlessly and were ensured for no repercussions based on the responses.

**Provenance and peer review** Not commissioned; externally peer reviewed.

**Data availability statement** Data may be obtained from a third party and are not publicly available. All data relevant to the study are included in the article or uploaded as supplementary information. All data generated during the study is included in the article. The data are not publicly available at present because it contains information that could compromise the confidentiality or consent of the study participants.

**ORCID iD**
Megha Sharma http://orcid.org/0000-0001-9165-9393

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
