## [Reviewer comments · BMJ Open]

ARTICLE DETAILS

TITLE (PROVISIONAL)	Impact of 'HIV-related stigma-reduction workshops' on knowledge and attitude of healthcare providers and students in Central India: A pre- and post-test intervention study
AUTHORS	Machowska, Anna; Bamboria, Babu Lal; Bercan, Courtney; Sharma, Megha

VERSION 1 – REVIEW

REVIEWER	Kathleen F. Norr University of Illinois at Chicago USA
REVIEW RETURNED	02-Sep-2019

GENERAL COMMENTS	Your work is very inspiring and needs to be published. In all the tables where you report the % for a yes-no question, please indicate that the answer is the % Yes. Your discussion feels too long, and I would recommend tightening so that it is not repetitive of results (in other words, don't repeat the results so much but just comment on them).
--

REVIEWER	Keivan Ahmadi University of Lincoln, the United Kingdom
REVIEW RETURNED	10-Nov-2019

GENERAL COMMENTS	Dear authors, I read with pleasure your manuscript titled: " Impact of HIV-related Stigma-reduction workshops on knowledge and attitude of healthcare providers and students in Central India: A pre- and post-test interventional study. Unfortunately, I could not find how your manuscript add anything new and substantial to the current extensive body of literature on HIV/AIDS-related stigma. My major concerns that have driven my decision are: 1- Methods: Post intervention is used carelessly and out of context. 2- Discussion: This section is out-dated and does not refer to the most recent/relevant publications (please refer to the Lancet Group publications) Minor comments: Interventional is a wrong word. I assume you meant " intervention".
--

REVIEWER	Barbara Friedland Population Council
-----------------	---

	USA
REVIEW RETURNED	06-Dec-2019

GENERAL COMMENTS	The need to educate healthcare providers about HIV/AIDS and to eliminate stigma and discrimination is indeed a critical issue. Unfortunately, in this reviewer's opinion, the study design in this paper was flawed because there was no evaluation of how well the intervention actually worked in terms of providers ACTIONS. The post-test questionnaire alone is not a valid measure of the success of the intervention -- by repeating the questionnaire right after the workshop, providers "know the right answers" to the questions. But the real evidence of whether or not the intervention worked requires them to put their learning into practice, which requires some longitudinal measure. If the authors had repeated the questionnaire to evaluate how the providers were incorporating what they had learned into service provision, they could then state that the workshop was effective. Furthermore, the authors make the erroneous conclusion that the length of the workshop is not relevant by comparing their results to a study of a 4-day workshop with different outcome measures. There have been a number of interventions with healthcare providers addressing stigma and discrimination amongst people living with HIV (see articles by Laura Nyblade) that involve more robust evaluations of the impact of the interventions. It may be possible for the authors to focus this paper solely on students and the benefits of incorporating a 2-hour workshop into existing curriculum. If presented in a more limited way, the paper might be more relevant to wider audiences.
---

VERSION 1 – AUTHOR RESPONSE

Reviewer: 1

1. *Your work is very inspiring and needs to be published. In all the tables where you report the % for a yes-no question, please indicate that the answer is the % Yes.*

Response: Thank you for your encouraging words and appreciation.

We have now indicated text as suggested by the reviewer.

2. *Your discussion feels too long, and I would recommend tightening so that it is not repetitive of results (in other words, don't repeat the results so much but just comment on them).*

Response: Thank you for the suggestion. We have now removed the duplication of results in discussion as suggested.

Reviewer: 2

Dear authors, I read with pleasure your manuscript titled: "Impact of HIV-related Stigma-reduction workshops on knowledge and attitude of healthcare providers and students in Central India: A pre- and post-test interventional study".

Unfortunately, I could not find how your manuscript add anything new and substantial to the current extensive body of literature on HIV/AIDS-related stigma.

My major concerns that have driven my decision are:

1. Methods:

Post intervention is used carelessly and out of context.

Response: Thank you for your comment. We are not sure if we understand this comment correctly, but we would like to emphasize that the data collection process was thorough, robust, transparent and reliable, and details on how the process followed are described in the methods section. We are fully aware of the limitations arising from pre/post intervention study design and we address them in our response to the reviewer 3.

2. Discussion:

This section is out-dated and does not refer to the most recent/relevant publications (please refer to the Lancet Group publications)

Response: Thank you for this comment and suggestion. The present manuscript was submitted in June 2018, by then the special issue of the Lancet was not published therefore, the references were not included. We have now included new references in the discussion as suggested and have changed the text accordingly to make it more relevant and updated.

Minor comments:

1. Interventional is a wrong word. I assume you meant "intervention".

Response:

Thank you for this suggestion. We have now changed the word to 'intervention' as suggested by the reviewer.

Reviewer: 3

1. The need to educate healthcare providers about HIV/AIDS and to eliminate stigma and discrimination is indeed a critical issue. Unfortunately, in this reviewer's opinion, the study design in this paper was flawed because there was no evaluation of how well the intervention actually worked in terms of providers ACTIONS.

The post-test questionnaire alone is not a valid measure of the success of the intervention -- by repeating the questionnaire right after the workshop, providers "know the right answers" to the questions. But the real evidence of whether or not the intervention worked requires them to put their learning into practise, which requires some longitudinal measure.

If the authors had repeated the questionnaire to evaluate how the providers were incorporating what they had learned into service provision, they could then state that the workshop was effective.

Response

Thank you for this comment. We agree that the practise of the current and future HCPs was not evaluated and would have strengthen the study. We also agree that the end-goal of any stigma reduction intervention should be reduced stigma behaviors, but the change in behaviors are difficult, and complex to capture and quantify at the workplaces, especially considering the study settings we were restricted to capture attitudes as a proxy. We had mentioned these points in the original version

of the manuscript. We have now highlighted this in text in the abstract, limitation and discussion sections.

Also, kindly note that the format of the intervention and intervention evaluation was tailored to the local context. It was first of its kind study performed in these settings (three colleges and two hospitals) aiming to include all present- and future-HCPs at various levels. Given the context specific conditions such as constrained resources for conducting and monitoring the longitudinal study, limited time of HCPs due to heavy workload, hesitance of some HCPs to be “examined” or “checked” for a sensitive topic of stigma and discrimination and finally, as emphasized in the literature, unawareness of the existing problem among HCPs, we took a pragmatic approach and adjusted the study design to all these factors.

As we stated in the results section “The implication of acquired knowledge and attitudes are complex, but theories suggest that knowledge is associated with baseline beliefs and attitudes and depends on the readiness of the learners and their willingness to integrate the change into their behavior”. Therefore, we truly believe that by capturing changes in knowledge and attitudes we provide enough evidence about the positive effect of the intervention.

We would further like to refer to the conceptual framework presented in the paper by Julie Pulerwitz, 2015, *Improving Hospital-Based Quality of Care by Reducing HIV-Related Stigma: Evaluation Results from Vietnam*, where it is presented that underlying causes of enacted stigma are fear-based stigma and value-based stigma.

We absolutely agree that the pre/post intervention study design comes with shortcomings and further repetitive follow-up studies would be needed to draw the conclusion that this specific intervention can change behavior in the setting. Therefore, we have now emphasized all methodological concerns in the limitation section to provide readers with a full picture.

2. Furthermore, the authors make the erroneous conclusion that the length of the workshop is not relevant by comparing their results to a study of a 4-day workshop with different outcome measures. There have been a number of interventions with healthcare providers addressing stigma and discrimination amongst people living with HIV (see articles by Laura Nyblade) that involve more robust evaluations of the impact of the interventions.

It may be possible for the authors to focus this paper solely on students and the benefits of incorporating a 2-hour workshop into existing curriculum. If presented in a more limited way, the paper might be more relevant to wider audiences.

Response

Thank you for this comment and suggestion. We have now added the text to stress out the importance of the benefits of incorporating a 2-hour workshop into existing curriculum

We would also like to emphasize that even though our intervention was much shorter than others, more elaborated interventions presented in the literature, all stigma reduction programs in healthcare facilities are urgently needed to improve the quality of care provided, uphold the human right to healthcare, increase access to health services, and maximize investments in HIV prevention and treatment.

Note: The manuscript has been carefully revised by a professional language editing service to enhance the grammar and legibility.